# Relationship between Metabolic Syndrome and Clinical Outcome in Patients Treated with Drug-Eluting Stenting after Rotational Atherectomy for Complex Calcified Coronary Lesions

**DOI:** 10.3390/jcm11144192

**Published:** 2022-07-19

**Authors:** Bin Hu, Changbo Xiao, Zhijian Wang, Dean Jia, Shiwei Yang, Shuo Jia, Guangyao Zhai, Hongya Han, Xiaohan Xu, Dongmei Shi, Yujie Zhou

**Affiliations:** 1Department of Cardiology, Beijing Anzhen Hospital, Capital Medical University, Beijing 100029, China; hubin80@126.com (B.H.); nongfu1975@126.com (Z.W.); jiadean120@163.com (D.J.); jackydang@163.com (S.Y.); jiashuo88@sina.com (S.J.); dzzhaiguangyao@163.com (G.Z.); hhy123100@163.com (H.H.); xiaoxiao640713@126.com (X.X.); 18910778615@163.com (D.S.); 2Beijing Institute of Heart, Lung and Blood Vessel Diseases, Beijing 100029, China; 3Department of Cardiovascular Surgery, Henan Chest Hospital, Zhengzhou 450001, China; xiaochb@163.com

**Keywords:** rotational atherectomy, metabolic syndrome, drug-eluting stent

## Abstract

Background and aims: although an association between metabolic syndrome (MS) and cardiovascular disease risk has been documented, the relationship in patients with complex calcified coronary lesions undergoing rotational atherectomy (RA) and drug-eluting stent(DES) insertion remains controversial. Here, the influence of MS on outcomes was assessed. Methods and results: we retrospectively included 398 patients who underwent RA and DES insertion for complex calcified coronary lesions in our institution between June 2015 and January 2019. The modified Adult Treatment Plan III was used to diagnose MS. The endpoint was major adverse cardiovascular events (MACEs), comprising mortality from all causes, myocardial infarction, and target vessel revascularization (TVR). In all, 173 (43.5%) patients had MS. MS was significantly associated with MACE over the 28.32 ± 6.79-month follow-up period (HR 1.783, 95% CI from 1.122 to 2.833) even after adjustment for other possible confounders. Conclusion: MS was frequently observed in patients treated with RA with DES insertion for complex calcified coronary lesions. MS independently predicted MACE in these patients.

## 1. Introduction

Calcification, mainly localized in the subintimal space, is a common phenomenon in the coronary arteries and is generally considered to be related to long-term arterial atherosclerosis [1,2]. Recent clinical and basic studies have shown that arterial calcification is an active, organized, and inflammation-dependent process, which is regulated by complicated cellular and enzymatic pathways and is possible to treat and prevent [3]. There has recently been much interest in the involvement of coronary artery calcium (CAC) in the process. CAC is closely related to age, sex, diabetes, hypertension, dyslipidemia, smoking, and metabolic syndrome (MS) [4,5], as well as elevated cardiovascular event risk [6,7].

Percutaneous coronary intervention (PCI) for calcified lesions, in contrast to non-calcified lesions, is frequently associated with poor clinical outcome and also represents a significant technical challenge for the clinician [8]. However, the use of rotational atherectomy (RA) makes it possible to modify the plaque associated with heavily calcified lesions and thus optimizes the preparation of the lesions [9].

MS leads to a significantly greater risk for the future development of both diabetes and cardiovascular diseases, together with increased mortality from all causes in a population initially free from these diseases. In addition, previous clinical investigations have shown MS was linked to poor prognosis in patients with coronary artery disease (CAD) [10,11,12,13,14,15]. Nevertheless, the associations between MS and clinical outcome in patients receiving RA and drug-eluting stent (DES) insertion for complex calcified lesions have not been determined. Here, the objective was, therefore, to examine these relationships.

## 2. Methods

### 2.1. Patient Data

The patients treated with drug-eluting stenting after RA for complex calcified coronary lesions at the Beijing Anzhen Hospital affiliated with the Capital Medical University between June 2015 and January 2019 were retrospectively enrolled. The baseline demographic and clinical features of the patients, including age, sex, risk factors for CAD, and medication, were obtained from the hospital databases. The patients had provided a sample of venous blood (after 24-h fasting) upon hospital admission. 

### 2.2. Definition of MS

Diagnosis of MS: MS was diagnosed following the modified criteria of the National Cholesterol Education Program Adult Treatment Panel III [16] as three of the following conditions: (1) obesity, defined as BMI values over 25 kg/m^2^ as there was no available information on waist circumferences; (2) high-density lipoprotein (HDL) values ≤1.29 mmol/L (women) or ≤1.03 mmol/L (men); (3) raised fasting plasma triglyceride (TG) levels (≥1.69 mmol/L), or on medication for raised TG; (4) systolic blood pressure ≥130 mmHg or diastolic value ≥85 mmHg, or receiving medication for hypertension (5) fasting plasma glucose (FPG) levels ≥6.1 mmol/L or diagnosis of diabetes mellitus.

### 2.3. Study Endpoint 

The study endpoint was major adverse cardiac events (MACE), including all-cause death, myocardial infarction (MI), and target vessel revascularization (TVR), either by PCI or coronary artery bypass grafting (CABG) during the follow-up. The dates and causes of death of patients who died at home were supplied by the families and the information was independently verified by clinicians.

### 2.4. Statistical Analysis 

Continuous variables were presented as means ± SDs and analyzed using *t*-tests. While categorical variables were presented in absolute numbers and percentages and χ^2^ tests were applied for these variables. Kaplan–Meier analysis, together with the log-rank test, were used for determining unadjusted rates. Multivariate Cox regression analysis was used for prognostic predictions, adjusting for probable confounders such as age, male sex, CAD family history, hypertension, smoking, diabetes mellitus (DM), dyslipidemia, chronic kidney disease (CKD), left ventricular ejection fraction (LVEF) <50%, previous MI, previous CABG, and medication. Two-tailed *p*-values < 0.05 represent statistical significance unless otherwise indicated. Analysis was conducted by SPSS version 22.0 (IBM Corp., Armonk, NY, USA).

## 3. Results

### 3.1. Study Design and Patients 

Three hundred and ninety-eight patients treated with drug-eluting stenting after RA for complex calcified coronary lesions in our institution between June 2015 and January 2019 were consecutively enrolled (Figure 1). The study was thus single-center and retrospective. Patients with severe anemia, active hemorrhage, or severe infection were not included, nor were those who were lost to follow-up. 

### 3.2. Baseline Characteristics 

Of the 398 patients, 173 (43.5%) had MS (the MS group) and 225 (56.5%) did not (the non-MS group) during hospitalization. Baseline features are provided in Table 1. Higher incidences of DM, stroke, hypertension, CKD, high BMI, high TG, low HDL-C, and high LDL-C were seen in the MS group. However, no differences were observed for current smoking, previous PCI, previous CABG, and left ventricular dysfunction.

The medications used during hospitalization included ACE inhibitors and hypoglycemic agents, with the former being most common in MS patients. No significant differences were seen for vasodilation and lipid-lowering therapy, as well as aspirin use, among the patients. 

### 3.3. Analysis of Mortality and MACE

One hundred and sixteen patients had one or more MACE during the follow-up of 28.32 ± 6.79 months. The events included mortality from all causes in 34 patients, MI in 22, and coronary revascularization in 68 (Table 2). Cumulative survival rates after the removal of MACE and TVR were markedly reduced in the MS group (Figure 1 and Figure 2) (log-rank test *p* < 0.001 (Figure 2); *p* < 0.001 (Figure 3)). However, Kaplan–Meier analysis indicated that the cumulative survival rates did not differ significantly for all-cause death and MI between the MS and non-MS groups (log-rank test, *p* = 0.060 for Figure 4 *p* = 0.232 for Figure 5).

Multivariate Cox analysis after exclusion of potential confounding factors, including age, male gender, current smoking, high TG, low HDL-C, high LDL-C, history of MI, CKD, previous CABG, BMI, stroke, LEVF < 40%, hypertension, DM, and medication, indicated that MS independently predicted MACE (HR 1.775; 95% CI 1.17–2.822; *p* = 0.015) and TVR(HR 2.658; 95% CI 1.390–3.080; *p* = 0.003) (Table 3). However, MS was not associated with all-cause death (HR 1.557; 95% CI 0.478–5.069; *p* = 0.462) and MI(HR 0.810; 95% CI 0.168–4.075; *p* = 0.798) (Table 3) during the follow-up.

## 4. Discussion

The essential findings of this retrospective investigation are: (1) there was a higher prevalence of MS in patients who had received RA and DES implantation for calcified coronary lesions; (2) and there was an association between MS and poor outcomes after removal of potentially confounding factors.

Heavily calcified coronary stenosis can result in failure of the stent to deliver adequately, possible polymer disruption, and either under-expansion or malapposition of the stent. It may also lead to a significant elevation in perioperative complications, as well as an increased likelihood of adverse events in the long term [17]. The RA procedure removes calcified atherosclerotic plaque effectively through the advancement of a high-speed diamond-encrusted elliptical burr through the artery. RA is also able to modify the plaque and enlarge the lumen of the artery, improving the conditions for the subsequent implantation and full expansion of the stent [9]. RA has been used in conjunction with all major types of PCI, namely, balloon angioplasty and the application of bare metal stents (BMS), and DES [18,19,20,21,22]. There is no link between RA and enhanced restenosis or repeat target lesion revascularization in complex calcified lesions in patients when used in conjunction with either balloon angioplasty or BMS [19,20,21]. Theoretically, the RA and DES combination can have a synergistic effect in complex calcified lesions due to the prevention of polymer damage afforded by RA and the effective inhibition of vascular neointimal proliferation by DES. Various studies have indicated improved long-term outcomes in patients who received RA and subsequent DES implantation, in comparison with RA and BMS [23,24]. This suggests that RA should be routinely used in combination with DES for patients with heavy CCS [25,26].

MS refers to a physiological and metabolic disorder that encompasses insulin resistance, elevated blood pressure, dyslipidemia, and abdominal obesity, and is frequently observed throughout the world [16]. The prevalence of MS varies according to the age group studied, race, geographical location, and different diagnostic criteria. However, it is estimated that at least one-quarter of adults suffer from the syndrome [27]. In China, the estimated prevalence among people over the age of 40 in 30 provinces was 34.0%, according to the revised ATP III criteria [28]. Most studies have shown that MS affects between 40% and 60% of CAD patients [13,14,29,30,31,32,33,34]. The present study was thus consistent in finding a 43.5% prevalence of MS in patients receiving RA and DES, although this was higher than that observed by the previous population survey.

Multiple clinical studies have found an increased association between MS and cardiovascular events in patients with established CAD [14,15,29]. In the GISSI-Prevenzione Trial [15], Levantesi et al. enrolled 11,323 patients with new onset MI and observed that MS patients had a markedly raised risk of MACE within the 3.5-year follow-up period. Younis et al. showed a strong association between MS and mortality from all causes in 15,524 CAD patients with stable disease over a 20-year follow-up period [13]. In addition, a systematic review of 55 studies with 162, 450 participants with cardiovascular disease observed a relationship between MS and increased relative risk (RR) for mortality from all causes (RR, 1.220; 95% CI, 1.103–1.349), death from cardiovascular disease (RR, 1.360; 95% CI, 1.152–1.606), MI (RR, 1.460, 95% CI 1.242–1.716), and stroke (RR, 1.435, 95% CI 1.131–1.820) compared with patients without MS [35]. Many studies have shown that poor outcomes are linked to MS in patients undergoing coronary revascularization [31,32,33,36,37,38]. The ten-year rates of cardiovascular events were observed to be elevated in MS patients after coronary artery bypass grafting [38], while Lovic et al. reported an independent association between MS and reduced overall survival in patients with acute MI after coronary stenting [31]. Hu et al. and Wang et al. also found an increased incidence of MACE in Chinese patients with MS after PCI [11,12]. An additional meta-analysis observed similar results in patients with CAD after coronary stenting [39].

In contrast, Zhou et al. reported comparable rates of MACE in MS and non-MS patients with primary PCI and multivessel coronary artery lesions [40]. Won and his colleagues also found that the presence of MS was not linked to long-term survival, although the incidence of DM, multivessel coronary artery lesions, history of PCI, and the mean number of DESs were considerably reduced in non-MS patients [34]. In addition, Patsa et al. reported that MS did not predict MACE in patients after new-generation DES implantation and showed a paradoxically favorable effect on clinical outcomes [41]. However, these authors only examined proximal left anterior descending artery lesions and recorded only five clinical events in 147 MS patients [41]. The discrepancies between these results may be a consequence of small sample sizes and differences in patient selection procedures, endpoints, observation periods, follow-up periods, or criteria.

Several investigations have shown MS was significantly linked to CAC and progression of CAD [5]. Furthermore, previous investigations showed that in-stent restenosis lesion, female gender, renal failure or hemodialysis, ACS at admission, procedural complications, and depressed LVEF were positively related to the increased occurrence of MACE after RA [42,43,44,45]. However, no studies have been undertaken on the effects of MS on the prognosis of patients receiving RA and DES insertion for complex calcified coronary lesions. This appears to be the first evaluation of the influence of MS on patient prognosis, indicating a significant link between MS and poor long-term clinical outcomes.

There were a few limitations to the present study. The first is that it was a non-randomized, single-center, retrospective investigation, which may lead to selection bias and reduced control for confounding factors. Second, the small sample size did not permit a comprehensive investigation. Third, due to the unavailability of data on waist circumferences, BMI was used instead. Nevertheless, it has been found that BMI ≥ 25.0 kg/m^2^ and central obesity (WC ≥ 90 cm for men or ≥80 cm for women) are in good accordance [46]. BMI has been used in many previous studies [13,34,36,40,41]. Lastly, we did not routinely determine the levels of cardiac enzymes after DES implantation. Thus, it is possible that the number of patients with periprocedural MI may have been underestimated. Large-sample, multicenter, prospective clinical studies or trials should be undertaken in the future.

## 5. Conclusions

Here, a significant association between MS and poor clinical outcomes was observed in patients receiving RA with DES insertion for complex calcified coronary lesions. These findings indicate that MS may be useful for evaluating risk and that early treatment of the disorder may enhance the outcomes of patients after RA and DES implantation for heavily calcified coronary lesions. 

## Figures and Tables

**Figure 1 jcm-11-04192-f001:**
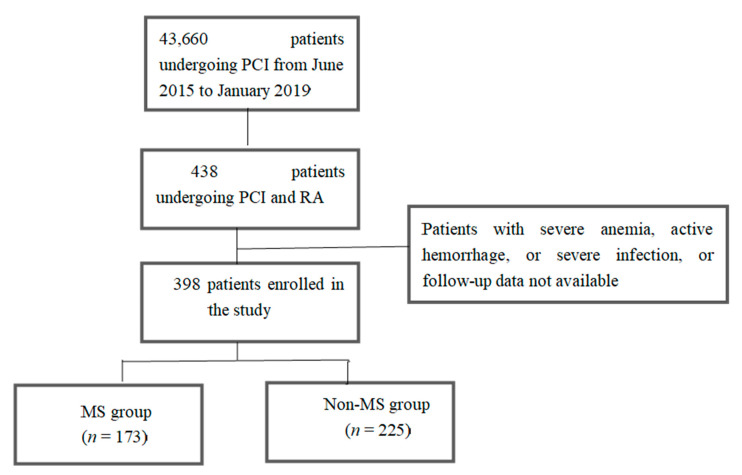
Study flow chart. 43,660 patients undergoing percutaneous coronary intervention (PCI) and 398 patients treated with treated with drug-eluting stenting after rotational atherectomy for complex calcified coronary lesions were consecutively enrolled between June 2015 and January 2019.

**Figure 2 jcm-11-04192-f002:**
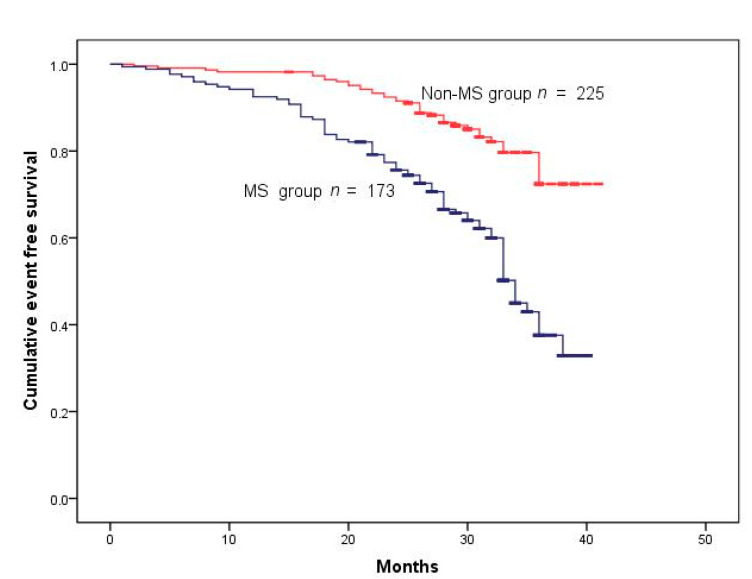
Cumulative survival rates censored for MACE. Survival was reduced in patients with MS (log-rank test *p* < 0.001).

**Figure 3 jcm-11-04192-f003:**
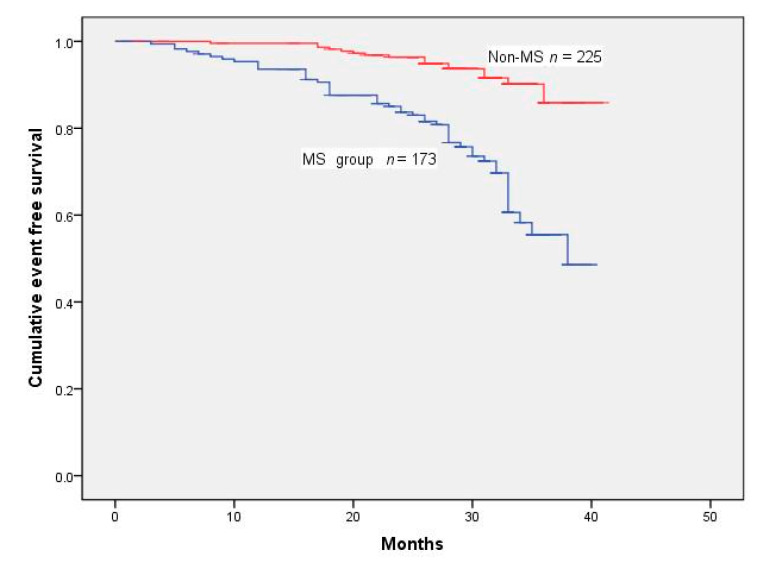
Cumulative survival rates censored for TVR. Survival was significantly reduced in MS patients (log-rank test *p* < 0.001).

**Figure 4 jcm-11-04192-f004:**
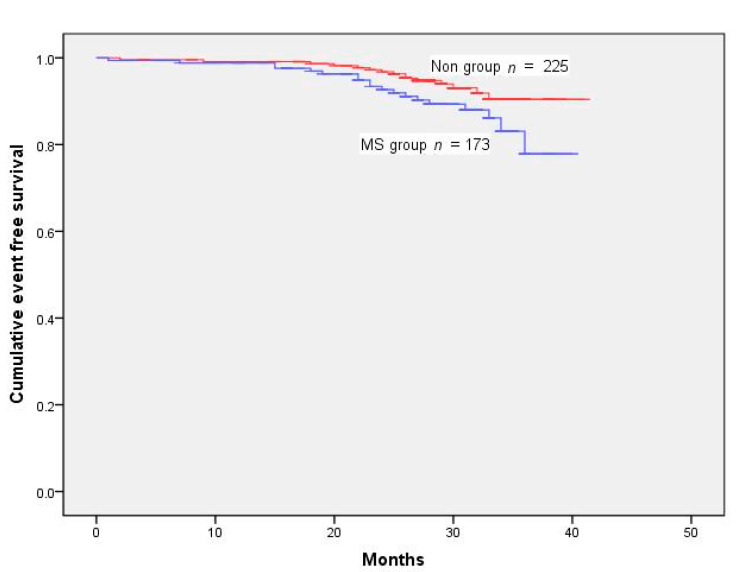
Cumulative survival rates censored for all-cause death. Survival rates were similar between the groups (log-rank test *p* = 0.060).

**Figure 5 jcm-11-04192-f005:**
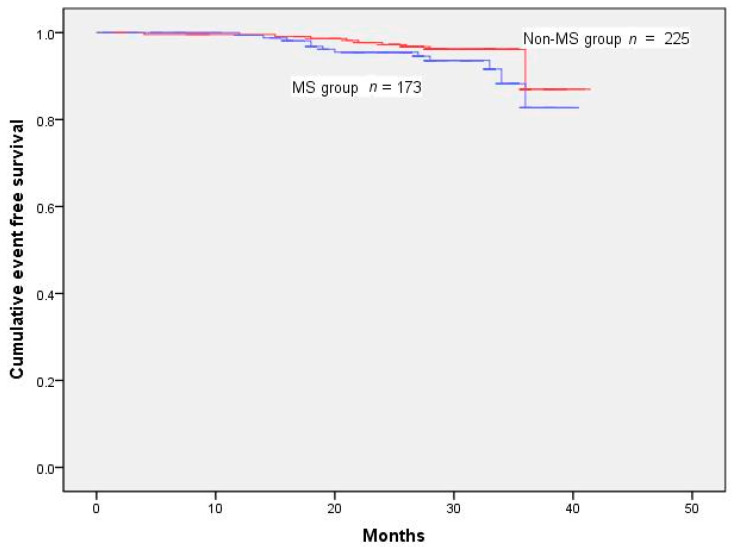
Cumulative survival rates censored for MI. Survival rates were similar between the groups (log-rank test *p* = 0.232).

**Table 1 jcm-11-04192-t001:** Patient characteristics at baseline.

	MS (*n* = 173)	No MS (*n* = 225)	*p*
Age (years) *	65.65 ± 8.664	65.27 ± 8.314	0.772
Male, *n* (%)	119 (68.79%)	149 (66.22%)	0.333
BMI (kg/m^2^) *	26.33 ± 1.75	24.59 ± 1.966	<0.001
DM, *n* (%)	119 (68.79%)	39 (17.33%)	<0.001
Hypertension, *n* (%)	154 (89.01%)	137 (60.89%)	<0.001
TG (mmol/L) *	2.292 ± 2.070	1.605 ± 1.042	<0.001
LDL-C (mmol /L) *	1.085 ± 0.297	1.206 ± 0.315	<0.001
HDL-C (mmol /L) *	2.99 ± 0.942	2.46 ± 0.729	<0.001
Current Smoking, *n* (%)	101 (58.38%)	153 (57.78%)	0.493
Family history of CAD, *n* (%)	22 (12.72%)	30 (13.33%)	0.490
Prior MI, *n* (%)	23 (13.29%)	26 (11.56%)	0.346
Stroke	33 (19.08%)	26 (11.56%)	0.026
Chronic Kidney failure, *n* (%)	46 (26.59%)	36 (16.00%)	0.007
Prior PCI	34 (19.65%)	37 (16.44%)	0.242
Previous CABG, *n* (%)	4 (2.31%)	5 (2.22%)	0.604
LVEF < 50%	7 (4.05%)	9 (4.00%)	0.588
Medication, *n* (%)			
Aspirin	171 (98.84)	222 (98.67%)	0.622
ACE inhibitors	144 (83.23%)	128 (56.89%)	<0.001
β-blockers	139 (80.35%)	173 (76.89%)	0.240
Statins	170 (98.27%)	222 (98.67%)	0.278
Nitrates	106 (61.27%)	154 (68.44%)	0.083
Hypoglycemic drug	109 (63.01%)	40 (17.78%)	<0.001

* Mean ± SD; ACE, angiotensin-converting enzyme; BMI, body mass index; CABG, coronary artery bypass graft; CAD, coronary artery disease; DM, diabetes mellitus; HDL-C, high-density lipoprotein cholesterol; LDL-C, low-density lipoprotein-cholesterol; LVEF, left ventricular ejection fraction; MI, myocardial infarction; MS, metabolic syndrome.

**Table 2 jcm-11-04192-t002:** Overall number of MACEs.

	MS (*n* = 173)	Non-MS (*n* = 225)	*p*-Value
MACE	78 (45.09%)	38 (16.89%)	<0.001
All-cause death	19 (10.98%)	15 (6.67%)	0.090
MI	12 (6.94%)	10 (4.44%)	0.195
TVR	51 (29.48%)	17 (7.56%)	<0.001

MACE, major adverse cardiovascular events; MI, myocardial infarction; MS, metabolic syndrome; TVR, target vessel revascularization.

**Table 3 jcm-11-04192-t003:** Prognostic value of MS *.

	HR	95% CI	*p*-Value
MACE	1.775	1.117–2.822	0.015
All-cause death	1.557	0.478–5.069	0.462
MI	0.810	0.168–4.075	0.798
TVR	2.658	1.390–5.080	0.003

* MACE, major adverse cardiovascular events; MI, myocardial infarction; MS, metabolic syndrome; TVR, target vessel revascularization; HR, hazard ratio; CI, confidence interval.

## Data Availability

The data presented in this study are available on request from the corresponding author. The data are not publicly available due to privacy.

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
