# Peer review of "Relationship between Metabolic Syndrome and Clinical Outcome in Patients Treated with Drug-Eluting Stenting after Rotational Atherectomy for Complex Calcified Coronary Lesions"

_jcm, 2022, doi:10.3390/jcm11144192_

Round 1

Reviewer 1 Report

The authors studied the “Relationship between metabolic syndrome and clinical out- 2 come in patients treated with drug-eluting stenting after rotational atherectomy for complex calcified coronary lesions”. Despite the many developments in understanding the metabolic syndrome in different clinical outcomes, there is still a necessity to further understand the role of MS in specific groups of patients with cardiovascular interventions. This article represents an important investigation, however below are some of the comments and concerns regarding the evaluation of this manuscript: 

1. Comments

  1. The authors introduced shortly their article. There is no main connection between the studies presented and the hypothesis or aims raised within the study.
  2. The authors did not introduce the role of metabolic syndrome on clinical outcomes after DES or chest pain in patients with CAD, for example (Hu Bin et al., 2011, PMID: 21422057, Patsa C et al., 2011, PMID: 21917434 [This is presented in discussion as reference 30], Kammerlander AA et al., 2021 PMID: 33558267; Wang HH et al, 2020, PMID: 32486596).
  3. A previously published meta-analysis also supports the role of MetS in MACE “Risk of major adverse cardiovascular events in patients with metabolic syndrome after revascularization: A meta-analysis of eighteen cohorts with 18,457 patients” however this study does not fully support this in terms of CABG or DES as mentioned by the author study investigation. How do authors comments on such findings from a higher number of patients and what is the main novelty of this retrospective study or comparison findings?
  4. What is the sample power calculation presented in your study?
  5. Patients' characteristics at the baseline have differences in ACE inhibitors, hypoglycemic drugs, and chronic kidney failure. How do authors comments on such difference within the baseline, and what is the impact on the main outcomes analyzed?
  6. The clear flow chart of patient selection is absent. Authors need to present a clear flow chart with criteria for selected patients. This is limited.
  7. The figure's quality is not so good!
  8. How did the authors evaluate or decide on the chosen confounding factors?
  9. What are the values of the complex coronary lesions analyzed in both groups of patients or CAC score?
  10. Authors did not discuss the comparison of similar observations within other groups than metabolic syndrome for example “Outcomes of rotational atherectomy for severely calcified coronary lesions: A single-center 5-year experience”  or “Comparison of Drug-Eluting Stent and Plain Old Balloon Angioplasty After Rotational Atherectomy in Severe Calcified and Large Coronary” Such studies made a clear analysis between the impact of DES and ROTA in the MACE and all cause mortality or even other diseases and its impact “Predictors of clinical outcome after rotational atherectomy-facilitated percutaneous coronary intervention in hemodialysis patients”
  11. Authors paved more attention to the metabolic syndrome, or introduced the incidences roles in cardiovascular effects and also the percentage within the population in China. This study is about having its impact on the worldwide population rather than being an epidemiological study with a focus on a specific population.
  12. The authors used a paragraph in the discussion “The discrepancies between these results may be a consequence of small sample sizes and differences in patient selection procedures, endpoints, observation periods, follow-up periods, or criteria” My question is what do the authors improve or add on within their study in terms of clarifying more such discrepancies, even though these are different in the design or topic of study.?
  13. The authors presented many limitations raised in the comments here as well, however regarding the BMI selection, one should think about the obesity paradox and its related impact in such observation. How do the authors think this might impact their study and since authors wanted to clarify between the clear Metabolic Syndrome?

Author Response

Thanks for giving us the opportunity to submit a revised draft of the manuscript “Relationship between metabolic syndrome and clinical outcome in patients treated with drug-eluting stenting after rotational atherectomy for complex calcified coronary lesions” for publication in Journal of Clinical Medicine. We feel great thanks for your professional review work on our article. As you are concerned, there are several problems that need to be addressed. According to your nice suggestions, we have made extensive corrections to our previous draft, the detailed corrections are listed below. Please see below, in red, for a point-by-point response to the reviewers’ comments and concerns. All page numbers refer to the revised manuscript file with tracked changes.

  1. The authors introduced shortly their article. There is no main connection between the studies presented and the hypothesis or aims raised within the study.

Thanks for your good suggestions. According to the comment, we added the connection between the studies presented and the hypothesis in the introduction. Page2, Line46-50.

  1. The authors did not introduce the role of metabolic syndrome on clinical outcomes after DES or chest pain in patients with CAD, for example (Hu Bin et al., 2011, PMID: 21422057, Patsa C et al., 2011, PMID: 21917434 [This is presented in discussion as reference 30], Kammerlander AA et al., 2021 PMID: 33558267; Wang HH et al, 2020, PMID: 32486596).

Thanks for your good suggestions. We sincerely appreciate the valuable comments. We have checked the literature carefully and added more references into the introduction and discussion part in the revised manuscript. Page2 Line50ï¼›Page8 Line 197-199ï¼›Page8 Line 208-209ï¼›Page8 Line 208-209ï¼›Page9 Line 276-281; Page10 Line 282-284;

  1. A previously published meta-analysis also supports the role of MetS in MACE “Risk of major adverse cardiovascular events in patients with metabolic syndrome after revascularization: A meta-analysis of eighteen cohorts with 18,457 patients” however this study does not fully support this in terms of CABG or DES as mentioned by the author study investigation. How do authors comments on such findings from a higher number of patients and what is the main novelty of this retrospective study or comparison findings?

Thanks for your good comments. The meta-analysis, included 18 studies with 18,457 patients showed that MS was linked to increased risks of MACE and all-cause mortality in patients after coronary revascularization, but not in patients receiving CABG or DES. However, most of the studies reported the clinical outcome of MACE. And only 8 of them included all-cause death. In addition, small sample sizes, different patient selection, shorted follow-up periods and different criteria of MS in these trials can affect the outcomes.

  1. What is the sample power calculation presented in your study?

Thanks for your good comments. The sample power calculation is important for clinical studies. This is a retrospective study enrolled the patients treated with drug-eluting stenting after rotational atherectomy for complex calcified coronary lesions between June 2015 and January 2019. And the sample sizes of the present study is larger than that of most studies about rotational atherectomy.

  1. Patients' characteristics at the baseline have differences in ACE inhibitors, hypoglycemic drugs, and chronic kidney failure. How do authors comments on such difference within the baseline, and what is the impact on the main outcomes analyzed?

Thanks for your good suggestions. The present study showed patients' characteristics at the baseline have differences in ACE inhibitors, hypoglycemic drugs, and chronic kidney failure. Multivariate Cox regression analysis was used for prognostic predictions, adjusting for probable confounders including medication. Page2,Line 85

  1. The clear flow chart of patient selection is absent. Authors need to present a clear flow chart with criteria for selected patients. This is limited.

Thanks for your good suggestions. According to the reviewer' good advice,we add figure 1 in the article. Page3,Line 94-99

  1. The figure's quality is not so good!

Thanks for your good suggestions. We have revised the figures.Page12-14, Line379-392.

  1. How did the authors evaluate or decide on the chosen confounding factors?

Thanks for your good suggestions. The present study was retrospective. And we collected the baseline data(including age, male sex, CAD family history, hypertension, smoking, diabetes mellitus , dyslipidemia, chronic kidney disease,left ventricular ejection fraction <50%, previous MI, previous CABG and medication) of patients as much as possible according to the previous studies.

  1. What are the values of the complex coronary lesions analyzed in both groups of patients or CAC score?

Thanks for your good suggestions. This is a retrospective study. And we can't obtained detailed information about the complex coronary lesions and CAC scores. Maybe we need more multicenter, prospective clinical trials to assess values of the complex coronary lesions and CAC score in the future.

  1. Authors did not discuss the comparison of similar observations within other groups than metabolic syndrome for example “Outcomes of rotational atherectomy for severely calcified coronary lesions: A single-center 5-year experience”  or “Comparison of Drug-Eluting Stent and Plain Old Balloon Angioplasty After Rotational Atherectomy in Severe Calcified and Large Coronary” Such studies made a clear analysis between the impact of DES and ROTA in the MACE and all cause mortality or even other diseases and its impact “Predictors of clinical outcome after rotational atherectomy-facilitated percutaneous coronary intervention in hemodialysis patients”

Thanks for your good suggestions. We added the comparison of similar observations similar observations within other groups than metabolic syndrome in the discussion part. Page8, Line215-218; Page8, Line215-218; Page11, Line366-369; Page12, Line370-376

  1. Authors paved more attention to the metabolic syndrome, or introduced the incidences roles in cardiovascular effects and also the percentage within the population in China. This study is about having its impact on the worldwide population rather than being an epidemiological study with a focus on a specific population.

Thanks for your good suggestions. The metabolic syndrome predispose an individual to a increased risk of developing type 2 diabetes and cardiovascular disease. Furthermore, the metabolic syndrome is also linked to poor prognosis in patients with coronary artery disease. We need large-sample, multicenter, prospective clinical studies to evaluate the impact of metabolic syndrome on the worldwide population  in the future.

12.The authors used a paragraph in the discussion “The discrepancies between these results may be a consequence of small sample sizes and differences in patient selection procedures, endpoints, observation periods, follow-up periods, or criteria” My question is what do the authors improve or add on within their study in terms of clarifying more such discrepancies, even though these are different in the design or topic of study?

Thanks for your nice comments. This was a non-randomized, single-center, retrospective investigation. Several investigations have shown MS was significantly linked to CAC and progression of CAD. And This present study appears to be the first trials of the influence of metabolic syndrome on the prognosis in patients treated with drug-eluting stenting after rotational atherectomy for complex calcified coronary lesions. Large-scale, prospective clinical investigations should be undertaken in the future to clarify such discrepancies.

  • The authors presented many limitations raised in the comments here as well, however regarding the BMI selection, one should think about the obesity paradox and its related impact in such observation. How do the authors think this might impact their study and since authors wanted to clarify between the clear Metabolic Syndrome?

         Thanks for your good suggestions. There was no available information on waist circumferences and we used BMI as a surrogate parameter for central obesity. Studies assessing the association between BMI and cardiovascular outcomes have produced conflicting findings(J  curve).Maybe because this is a small sample sizes and retrospective study, we did not find the association between BMI and the clinical outcomes in  patients treated with drug-eluting stenting after rotational atherectomy for complex calcified coronary lesions. And the objective of the present study was to examine these relationships between Metabolic Syndrome and the clinical outcomes in those patients. Therefore, we did not list the result in the article.

Reviewer 2 Report

The Discussion section has been very well described in detail but maybe if possible, could be reduced a little. Separate paragraph specifying significance of the study if incorporated would be helpful. The authors could include the novelty of their study within a separate“significance” section

Author Response

Thank you again for your positive comments and valuable suggestions to improve the quality of our manuscript. According to your good suggestions, we have made corrections to our previous draft, the detailed corrections are listed below.

The Discussion section has been very well described in detail but maybe if possible, could be reduced a little. Separate paragraph specifying significance of the study if incorporated would be helpful. The authors could include the novelty of their study within a separate“significance” section

 Thanks for your good suggestions. .According to the reviewers’comments, we revised the manuscript and reduced the Discussion section and add the specifying significance of the study.Page8,Line213-217.

Reviewer 3 Report

In this paper, Bin Hu et al found that MS was frequently observed in patients treated with RA with DES insertion for complex calcified coronary lesions and that MS was independently associated with MACE.

Major comments:

1.    Authors should specify references for the first two sentences of the Introduction section;

2.    The sentence on page 2, line 50 should go into the Results not the Methods section;

3.    The follow-up time (28.32±6.78 months) should go into the Results not the Methods section;

4.    In the "Statistical analysis section" it must be specified that the categorical variables have been reported in absolute numbers and percentages;

5.    Authors should specify references for the sentence in lines 151-153 of the Discussion section;

6.    The way of writing in English must be reviewed throughout the article, preferably by a native speaker;

7.    There is no trace as to whether or not an approval from the ethics committee was obtained, nor if informed consent was acquired (even if the study was retrospective): please, provide a clarification in the text.

Minor comments

1.    Leave space between words and references in square brackets;

2.    After the numerical list and the round bracket, the period does not go too;

3.    BMI in the text has no unit of measurement (kg / m2);

4.    "mm Hg" must be changed in the text to "mmHg";

5.    “mmol/l” must be changed in the text to “mmol/L”;

6.    The formatting of Table 1 must be reviewed and made uniform with the text;

7.    In the caption of Table 1, you should change “angiotensin converting enzyme” to “angiotensin-converting enzyme”;

8.    The formatting of Table 3 must be reviewed and made uniform with the text;

In conclusion, I think that the work needs to be improved a lot, not only in the exposition in English but also in the order of organization of the various sections.

Author Response

   Thanks for your good comments. We have carefully reviewed the comments and have revised the manuscript accordingly. If there are any other modifications we could make, we would like very much to modify them and we really appreciate your help. Our responses are given in a point-by-point manner below. Changes to the manuscript are shown in red.We hope that our manuscript could be considered for publication in your journal. Thank you very much for your help.

Major comments:

  1. Authors should specify references for the first two sentences of the Introduction section;

Thank you for your good suggestions. According to the reviewers’ good comments, we have added the references in the introduction section. Page1.Line34,37; Page9.Line249-260

  1. The sentence on page 2, line 50 should go into the Results not the Methods section;

Thank you for your good suggestions. According to the reviewers’ good comments, we have added the references in the introduction section. Page2,Line89; Page3,Line90-94

  1. The follow-up time (28.32±6.78 months) should go into the Results not the Methods section;

Thanks for your good suggestions. According to the comment, the follow-up time (28.32±6.78 months) have gone into the Results section. Page4,Line117

  1. In the "Statistical analysis section" it must be specified that the categorical variables have been reported in absolute numbers and percentages;

Thanks for your good suggestions. According to the reviewers’ good suggestions,we have added “categorical variables were presented in absolute numbers and percentages ”in the Statistical analysis section. Page2,Line79

  1. Authors should specify references for the sentence in lines 151-153 of the Discussion section;

Thanks for your good comments. According to the reviewers’ good suggestions, we have added the references in the Discussion section. Page7,Line163,165;Page10,Line299-312

  1. The way of writing in English must be reviewed throughout the article, preferably by a native speaker;

Thanks for your good suggestions. According to the reviewers’ good suggestions,we have a native speaker reviewed English writing of the article.

  1. There is no trace as to whether or not an approval from the ethics committee was obtained, nor if informed consent was acquired (even if the study was retrospective): please, provide a clarification in the text.

 Thanks for your good suggestions. According to the reviewers’ good suggestions, we have added the section of the section of "Ethics approval and consent to participate".Page9,Line238-240

Minor comments

  1. Leave space between words and references in square brackets;

Thanks for your good suggestions. According to the reviewers’ good suggestions, we have revised them.

  1. After the numerical list and the round bracket, the period does not go too;

Thanks for your good suggestions. According to the reviewers’ good suggestions,we have changed the period.

  1. BMI in the text has no unit of measurement (kg / m2);

Thanks for your good suggestions. According to the reviewers’ good suggestions, we have added the unit of BMI. Page2 Line65.

  1. "mm Hg" must be changed in the text to "mmHg";

Thanks for your good suggestions. According to the reviewers’ good suggestions,"mm Hg" has be changed to "mmHg" in the text. Page2 Line65.Tabel1

  1. “mmol/l” must be changed in the text to “mmol/L”;

Thanks for your good suggestions. According to the reviewers’ good suggestions, “mmol/l” has be changed to to “mmol/L” in the text. Page2 Line67-70.Table1’

  1. The formatting of Table 1 must be reviewed and made uniform with the text;

Thanks for your good suggestions. According to the reviewers’ good suggestions, we have revised the Table 1.

  1. In the caption of Table 1, you should change “angiotensin converting enzyme” to “angiotensin-converting enzyme”;

Thanks for your good suggestions. According to the reviewers’ good suggestions,we have changed “angiotensin converting enzyme” to “angiotensin-converting enzyme”. Page4 Line107

  1. The formatting of Table 3 must be reviewed and made uniform with the text;

 Thanks for your good suggestions. According to the reviewers’ good suggestions ,we have revised the Table 3.

Round 2

Reviewer 3 Report

The authors addressed all the questions.

Author Response

Thanks again for your positive comments on our manuscript and giving us the opportunity to submit a revised draft of the manuscript “Relationship between metabolic syndrome and clinical outcome in patients treated with drug-eluting stenting after rotational atherectomy for complex calcified coronary lesions(JCM- 1792895)” for publication in the Journal of Clinical Medicine. We feel great thanks for your professional review work on our article. As you are concerned, there are several problems that need to be addressed. According to your nice suggestions, we have made extensive corrections to our previous draft, the detailed corrections are listed below. Please see below, in red, for a point-by-point response to the reviewers’ comments and concerns. All page numbers refer to the revised manuscript file with tracked changes.